# Secondhand Smoke Exposure and Its Impact on Pediatric Lung Function, Aerobic Fitness, and Body Mass: Evidence from a Cross-Sectional Study

**DOI:** 10.3390/children11101250

**Published:** 2024-10-17

**Authors:** Ivan Pavić, Iva Topalušić, Tamara Poljičanin, Ozana Hofmann Jaeger, Sara Žaja, Asja Stipić Marković

**Affiliations:** 1Department of Pulmonology, Allergology, Immunology and Rheumatology, Children’s Hospital Zagreb, Klaićeva Street 16, 10 000 Zagreb, Croatia; ipavic01@gmail.com (I.P.); ozanahj@gmail.com (O.H.J.); 2School of Medicine, University of Split, Šoltanska 2, 21 000 Split, Croatia; sara01zaja@gmail.com; 3Zagreb County Health Center, Josip Runjanin Street 4, 10 000 Zagreb, Croatia; tamara.poljicanin@gmail.com; 4University Hospital for Infectious Diseases Dr. Fran Mihaljević, Mirogojska 8, 10 000 Zagreb, Croatia; asjastipic90@gmail.com

**Keywords:** secondhand smoke, lung function, physical fitness, children

## Abstract

Background: Several studies have documented the detrimental impacts of secondhand smoke (SHS) exposure to a range of pediatric respiratory conditions, including asthma, bronchitis, and reduced lung function. The aim of the study was to investigate the influence of SHS exposure on lung function, physical fitness, and body mass index (BMI) in children aged 10 to 14 years. Methods: This cross-sectional study included children aged 10 to 14 years at the Elementary School “Trilj” in Trilj, Croatia. Data on SHS exposure were collected using a questionnaire. Antropometric and spirometry measurements were performed. Physical fitness was assessed using the shuttle run (BEEP) test. Results: This study included 157 children, 89 (56.69%) boys and 68 (43.31%) girls. Children exposed to every day SHS in households had significantly lower values of forced vital capacity (FVC), forced expiratory volume in one second (FEV1), FEV1/FVC, peak expiratory flow (PEF) (*p* < 0.001) and higher z-score BMI levels (*p* = 0.018) in comparison to unexposed children. Logistic regression showed that children unexposed to SHS had higher odds for better results in the BEEP test (OR 62.45, 95% CI 21.26–179.24, *p* < 0.001). Children with poorer physical fitness, expressed by lower BEEP score levels, had significantly lower FVC, FEV1, FEV1/FVC, and PEF (*p* < 0.001). Conclusions: Every day SHS exposure in children was associated with poorer lung function, higher BMI, and poorer physical fitness.

## 1. Introduction

Tobacco smoking is a pervasive global health concern, contributing significantly to the burden of respiratory diseases and posing a substantial risk to both smokers and those exposed to secondhand smoke (SHS). The adverse health effects of active smoking on adult lung function are well-documented [1,2]. However, the impact of parental smoking on the respiratory health of their offspring is an area that warrants further investigation. Exposure to SHS remains a significant public health issue, particularly for children, who are more vulnerable to its harmful effects due to their developing respiratory and immune systems [3,4,5]. Despite widespread public awareness campaigns and smoking restrictions in many countries, parental smoking continues to be a common source of SHS exposure for children [3,4,5]. Numerous studies have documented the detrimental impacts of SHS on pediatric health, linking it to a range of respiratory conditions, including asthma, bronchitis, and reduced lung function [6,7,8,9,10]. One critical area of concern is the influence of parental smoking on the lung function and overall physical fitness of children. Spirometry, a non-invasive and widely used method to assess lung function, provides valuable insights into the respiratory health of children. Parameters such as forced vital capacity (FVC), forced expiratory volume in one second (FEV1), and the ratio of FEV1 to FVC (FEV1/FVC), are essential for evaluating both the overall respiratory health of individuals and the impact of SHS on pulmonary function [11].

In addition to lung function, physical fitness, particularly aerobic capacity, is an important indicator of overall health in children. The BEEP test, also known as the shuttle run test, is a widely recognized method for assessing aerobic fitness and functional abilities [12]. It measures the maximum oxygen uptake (VO_2_ max) and endurance capacity, providing a comprehensive evaluation of a child’s physical fitness. This study aims to investigate the influence of parental smoking on the functional abilities and lung function of children aged 10 to 14 years. By using spirometry to assess lung function and the BEEP test to evaluate physical fitness, we seek to understand the extent to which SHS exposure from parental smoking affects these crucial health parameters. Specifically, we compared the spirometry measurements and BEEP test results between children exposed to parental smoking and those from non-smoking households. By doing so, we aim to highlight the potential health risks associated with parental smoking and underscore the importance of reducing SHS exposure in homes. Understanding the impact of parental smoking on children’s health can inform public health policies and interventions aimed at protecting this vulnerable population. By providing empirical evidence on the adverse effects of SHS, this research can contribute to the ongoing efforts to promote smoke-free environments for children and improve their long-term health outcomes.

## 2. Materials and Methods

### 2.1. Study Population

This cross-sectional study included children aged 10 to 14 years. The study was conducted at the Elementary School “Trilj” in Trilj, Croatia, between November 2023 and June 2024. Trilj is a small town with a population of approximately 9000 citizens, located in the Dalmatian hinterland. The region is characterized by its rural setting and a relatively homogenous socio-economic structure, with minimal differences in income levels and living standards among its inhabitants. This socio-economic uniformity helps to minimize potential confounding factors related to economic disparities in the study. This study was conducted in accordance with the Declaration of Helsinki and was approved by the Ethics Committee of the Children’s Hospital Zagreb. Written consent was obtained from parents and children.

### 2.2. Inclusion and Exclusion Criteria

#### 2.2.1. The Inclusion Criteria Included the Following

Children between 10 and 14 years;Only children who were in good overall health, with no acute or chronic illnesses affecting respiratory or systemic health, were included;Children must have had no known respiratory infections for at least one month prior to testing to ensure accurate respiratory function measurements.

#### 2.2.2. The Exclusion Criteria Included the Following

Children diagnosed with chronic respiratory conditions or other chronic diseases;Children with any acute respiratory or systemic illness at the time of the study or within the one-month period prior to testing;Presence of systemic illnesses that could affect the respiratory system (e.g., cardiovascular diseases, neuromuscular disorders, immunodeficiency disorders);Children on medications that could affect respiratory function, such as bronchodilators or corticosteroids;Children whose parental smoking status could not be clearly determined through the questionnaire.

By adhering to these criteria, the study ensured the inclusion of healthy children without recent respiratory infections, thus providing reliable data on the impact of parental smoking on children’s respiratory health and physical fitness.

### 2.3. Data Collection

Information on exposure to cigarette smoke was collected using a self-created questionnaire, answered by parents and children. The questionnaire consisted of 14 questions. In the first part, data on age, sex, and chronic respiratory conditions were collected including any acute respiratory or systemic illness at the time of the study or within the one-month period prior to testing, asthma, allergic rhinitis, presence of systemic illnesses that could affect the respiratory system (cardiovascular diseases, neuromuscular disorders, immunodeficiency disorders), or medications that could affect respiratory function, such as bronchodilatators or corticosteroids. In the second part, the data on the SHS exposure were collected, including exposure to the secondhand smoke at home (yes/no), current mother’s/father’s smoking or both and mother’s smoking during pregnancy. Additional information regrading active sport was obtained (yes/no).

Based on parental smoking habits, the children were divided into two groups as follows:

Group I: Children of smoking parents (children whose one or both parents smoke every day in the family environment);

Group II: Children of non-smoking parents (control group).

### 2.4. Anthropometric Measurements

Weight and height were measured using a digital electronic weighing scale Seca^®^ (range 1–150 kg) and a digital stadiometer (range 70–205 cm) (seca GmbH & Co. KG., Hamburg, Germany) which were calibrated. Weight was recorded in kilograms (kg) and height in centimeters (cm). These measurements were used to calculate each subject’s body mass index (BMI), which was then expressed as a BMI z-score.

### 2.5. Spirometry Measurements

Spirometry was performed between 8 AM and 10 AM using a portable spirometer (Carefusion Germany 234 GmbH, Höchberg, Germany). The following parameters were measured: forced vital capacity (FVC), forced expiratory volume in 1 s (FEV1), FEV1/FVC ratio, peak expiratory flow (PEF).

Children were instructed to perform the test according to standardized procedures [11]. They were asked to take a deep breath and then exhale as forcefully and quickly as possible into the spirometer. Each child performed three acceptable maneuvers, and the highest values of FVC and FEV1 were recorded. The results were compared to reference values adjusted for age, sex, height, and ethnicity.

### 2.6. Assessment of Motor Skills

Motor skills were assessed using the BEEP test, which evaluates cardiovascular endurance and physical fitness [12]. The BEEP test involves running back and forth across a 20 m distance at increasing speeds, dictated by audio signals. The test starts at a slow pace, and the speed increases incrementally each minute. Children continue running until they can no longer keep up with the set pace. The original data in the research paper were presented as percentiles [12]. These were converted to a more easily understood rating value. The conversion used was <P5 = very poor (score 1), P5–P20 = poor (score 2), P20–P40 = fair (score 3), P40–P60 = average (score 4), P60–P80 = good (score 5), P80–P95 = very good (score 6), >P95 = excellent (score 7). For the purposes of certain analyses, we combined scores 1 with 2 and 6 with 7 (<P5 = very poor combined with P5–P20 = poor, and P80–P95 = very good combined with >P95 = excellent).

### 2.7. Statistical Analyses

Descriptive statistics analyses were performed for all investigated variables. Normality of distribution was tested using Shapiro–Wilks test, while homogeneity of variance was tested using Levene test. Data were presented as mean ± standard deviation in the case of normal distribution and median, minimum, and maximum when there was a violation of normality present. Categorical data were presented as frequencies and percentages. In order to minimize sex and age bias, z-scores were calculated for BMI and FEV1/FVC. Differences between groups of independent continuous variables were analyzed using student *t*-test test and Kruskal–Wallis test. Differences in the prevalence of individual conditions were compared using the χ2-test and Fisher’s Exact Test. The significance of the correlation between variables were analyzed by Spearman’s rank order correlations test.

Ordinal logistic regression was performed for the prediction of the probability of BEEP score and FEV1/FVC ratio in relation to SHS exposure and BMI z-score, with the strength of association quantified by odds ratio (OR) and 95% confidence interval (CI). An error threshold of α = 0.05 was used in the interpretation of the results. All statistical analyses were performed using SPSS software (26.0, SPSS Inc., Chicago, IL, USA).

## 3. Results

### 3.1. Demographic Data

The study included 157 elementary school pupils in the Sinj region. The average age ± standard deviation (SD) was 12.55 ± 1.05 years. There were 89 (56.69%) boys and 68 (43.31%) girls. The baseline characteristics of investigated population are presented in Table 1.

Among 98 (62.4%) pupils exposed to SHS, 75 (76.5%) of them had fathers and 79 of them (80.6%) had mothers who smoked in homes every day. Furthermore, 42 (26.7%) pupils in total had one parent and 56 pupils (35.7%) had both parents who smoked daily. During pregnancy, 60 (61.2%) pupils were exposed to SHS and all of them were exposed to SHS from a mother afterwards.

### 3.2. Association of SHS Exposure, Lung Function, and BMI

In children exposed to every day SHS in households, significantly lower values of FVC, FEV1, FEV1/FVC, and PEF were detected in comparison to unexposed children (*p* < 0.001). Significantly lower values of FVC, FEV1, FEV1/FVC, and PEF were found as well in children who had either a father or a mother that smoked, and in children exposed to SHS during pregnancy (*p* < 0.018, 0.046, 0.001, 0.016). No statistically significant difference in FVC, FEV1, FEV1/FVC, and PEF was observed between boys and girls (*p* = 0.694, *p* = 0.347, *p* = 0.572, *p* = 0.134), or between children with and without self-reported active sport status (*p* = 0.178, *p* = 0.112, *p* = 0.368, *p* = 0.384) (Table 2).

BMI did not differ significantly between boys and girls (*p* = 0.441) and those with active sport status (*p* = 0.601) either. However, children exposed to SHS had statistically higher z-score BMI levels (*p* = 0.018). Significantly higher z-score BMI levels were found also in children who had either a father or a mother that smoked (*p* = 0.046, *p* = 0.011) and in children exposed to passive smoking during pregnancy (*p* = 0.009) (Table 2).

Children who were not exposed to SHS had higher odds for better results in BEEP tests, measured by higher BEEP score levels (OR 62.45, 95% CI 21.26–179.24, *p* < 0.001). There was no significant difference with regards to FEV1/FVC (Tiffeneau z-score) (OR 1.07, 95% CI 0.74–1.55, *p* = 0.707). The association between passive smoking exposure and BEEP test levels are presented in Figure 1.

In addition, an ordinal logistic regression model for the outcome of BEEP score level revealed that an increase in BMI (z-score) was associated with lower the odds for better BEEP score level, with an OR of 0.37 (95% CI, 0.26 to 0.51), *p* < 0.001.

We also investigated the association of between physical fitness and lung function. Children with poorer physical fitness, expressed by lower BEEP score levels, had significantly lower FVC, FEV1, FEV1/FVC, and PEF (all *p*’s < 0.001), as shown in Table 3. Children with lower BEEP score levels had significantly higher BMI (*p* < 0.001) (Table 3).

## 4. Discussion

The results of our study indicate a significant association between SHS exposure, BMI, and impaired lung function, as well as reduced physical fitness, measured by BEEP test scores. These findings align with existing literature, reinforcing the detrimental impacts of SHS on pediatric respiratory health and overall physical fitness [7,8].

Our study demonstrated significantly lower values of FVC, FEV1, FEV1/FVC ratio, and PEF among children exposed to SHS compared to their non-exposed peers. These findings are consistent with the well-documented adverse effects of SHS on respiratory health [8,9,10,11,12,13,14,15,16].

Previously, Li YF et al. analyzed the effects of in utero and SHS exposure on lung function in children with and without asthma across 12 study communities, involving 5263 students and reported a 9% reduction in FEV1 among children exposed to passive smoking [15]. More recently, the study by Milanzi et al. using data from 552 participants in the Dutch PIAMA cohort, found that SHS exposure during childhood was associated with reduced lung function growth and lower attained lung function in adolescence [16]. Our study corroborates the findings of both cited studies, showing significantly lower values of FVC, FEV1, FEV1/FVC, and PEF in children exposed to passive smoking. This consistent evidence across studies underscores the detrimental impact of passive smoking on lung function, emphasizing a substantial decline in respiratory capacity and efficiency among exposed children [17,18]. In a prospective birth cohort of Tucher JD et al., maternal smoking during pregnancy was associated with lower FEV_1_/FVC ratios in the offspring and increased airway resistance at the age of 16 [18]. However, Thucher JD et al. did not find significant associations between exposure to SHS during infancy or adolescence and reduced FEV1 or FVC values [18]. SHS exposure was related to an increase in FVC in this study, leading to the decrease in FEV1/FVC ratio. The discrepancies in findings between our study and Thucher JD et al. can likely be attributed to a combination of demographic, methodological, exposure-related, genetic, environmental, and lifestyle factors. For instance, higher levels of air pollution (e.g., ozone level) had an additive effect to the decrease in lung function in children exposed to SHS [19]. In the study of 1819 participants aged 8–17 years, a decrease in FEV1 and FVC was observed according to the number of persons smoking at home. It has been known that cigarette smoke consists of thousands of irritative, toxic, and pharmacologically active substances. The main mechanisms of SHS leading to lung function deterioration are inflammatory reaction caused by irritants, enhanced resistance of the airways, and more frequent respiratory infections [20,21,22]. Understanding these differences is crucial for interpreting the results and for designing future research that can better isolate and identify the specific impacts of smoking exposure on children’s health.

The decreased spirometry values in our study were detected also in children with SHS exposure during pregnancy and with mothers who smoked afterwards, which highlights the pervasive impact of SHS on the respiratory health of children. In our study, a high proportion of mothers smoked during pregnancy, and a similar proportion was found in another region of Croatia [23]. As in other studies, most of the mothers who smoked during pregnancy continued to smoke after the child’s birth [17]. Prenatal SHS exposure is particularly associated with a decrease in lung function [24]. A study of 20,000 children aged 6–12 years from 9 European and North American schools showed a decrease of −1% FEV1 in children exposed to SHS in-utero, while a decrease was weaker in postnatally exposed children (−0.5% FEV1) [24]. Prenatal SHS exposure can lead to the decreased lung growth and higher frequency of respiratory infections [19]. This reduction in lung function can predispose children to a range of respiratory conditions, including asthma and chronic bronchitis, further underlining the importance of reducing SHS exposure in homes [8,13,19].

In addition to lung function, our study assessed physical fitness using the BEEP test, a widely recognized measure of aerobic capacity. We found that children exposed to SHS had significantly lower BEEP test scores, indicating poorer physical fitness compared to non-exposed children. This result is particularly concerning as physical fitness is a crucial indicator of overall health and well-being in children. The association between SHS exposure and reduced physical fitness has been less frequently studied compared to lung function; however, our findings contribute to the growing body of evidence suggesting that SHS can adversely affect physical development and performance [25,26,27]. The ordinal logistic regression model revealed that SHS exposure significantly decreased the odds of achieving higher BEEP test scores, with an odds ratio of 62.45, suggesting a strong negative impact of SHS on children’s aerobic capacity. This is in line with Erkelenz et al., who emphasized the broader health implications of household smoking, including its potential impact on physical activity and fitness [26,28].

The relationship between SHS exposure and increased BMI in our study is also noteworthy. Children exposed to SHS had a significantly higher BMI compared to non-exposed children. This finding is supported by other studies suggesting that SHS exposure may be associated with obesity in children, possibly due to the influence of smoking on metabolic and endocrine functions [29,30,31,32]. Recently, Jaakkola et al. found that childhood exposure to parental smoking was linked to a higher risk of developing overweight status and obesity throughout life [29]. Specifically, the Young Finns Study reported a 13% increased risk of overweight/obesity and an 18% increased risk of central obesity, while the Special Turku Coronary Risk Factor Intervention Project between years 1989 and 2009/2010 study found a 57% increased risk of overweight/obesity and a 45% increased risk of central obesity [30,31]. These findings are consistent with the results of this study, which also demonstrated a significant association between parental smoking and increased risks of overweight/obesity.

The link between SHS exposure and increased BMI in children can be attributed to several factors. Nicotine and other chemicals in tobacco smoke can interfere with metabolic processes, leading to weight gain [25,33,34,35,36]. SHS exposure has been shown to alter hormone levels, such as leptin and insulin, which regulate appetite and energy balance, thereby promoting weight gain and obesity in children [35]. Additionally, parental smoking is often associated with less healthy household environments and behaviors, including poorer diet quality and reduced physical activity opportunities for children [36]. SHS exposure was associated with longer time spent in front of the screen and inadequate sleep in Spanish children [25]. This combination of direct physiological effects and lifestyle factors contributes to the higher prevalence of overweight and obesity observed in children exposed to SHS.

Data on the association between physical fitness and lung function are limited, and the results of the studies are mixed [37]. Hancox and Rasmussen investigated associations between lung function and fitness in two population-based cohort studies of children and young adults and showed that aerobic fitness was associated with higher values of FEV1 and FVC among children, adolescents, and young adults, but not in FEV1/FVC ratio, suggesting that aerobic fitness could influence lung volumes rather than airway caliber [37]. In our study, physical fitness, expressed as a better level of the BEEP test, was also associated with higher FEV1, FVC, but also FEV1/FVC ratio. Interestingly, our study did not find significant differences in lung function between children who reported being active in sports and those who were not, regardless of their exposure to SHS. This lack of association might be explained by the fact that children who engage in regular sports might develop compensatory mechanisms that help maintain their lung function and fitness levels despite SHS exposure. However, these mechanisms might not be strong enough to show significant statistical differences in this study. On the other hand, the benefits of physical activity might manifest more prominently over a longer period, as shown in the study of Hancox et al. [37]. The cross-sectional nature of our study captures a snapshot in time and might not fully reflect the long-term protective effects of physical activity against SHS exposure. Finally, the classification of children into “active sportspersons” might include a wide range of activity levels, from occasional participation to regular intense training. This variation could dilute the potential differences that might be more apparent in a more homogeneously active group.

### Limitations of the Study and Recommendations for Future Research

While our study provides valuable insights, it is not without limitations. The cross-sectional design limits our ability to establish causal relationships. Additionally, reliance on self-reported data for SHS exposure and physical activity may introduce reporting biases. Self-reported exposure to SHS can lead to mistakes in classifying who is exposed or not. However, we believe the use of self-reported data is still a valid approach for assessing exposure, as evidenced by findings from existing literature. Becher and colleagues conducted an extensive study comparing passive smoking exposure between Germany and Poland, where they found that the validity of self-reported exposure to passive smoking was generally good [38]. Their findings indicate that self-reports are reasonably accurate, suggesting they can be used as an acceptable alternative when biomarker analysis is not feasible. Furthermore, a large-scale international study involving ten countries demonstrated that self-reported measures, such as the duration of exposure and the number of cigarettes reported, were strongly correlated with urinary cotinine levels [39]. The study by Jurado D et al. demonstrated a significant association between parental smoking behavior, as reported in questionnaires, and the levels of urinary cotinine in children [40]. This supports the idea that questionnaire data can effectively estimate exposure to environmental tobacco smoke in a reliable manner. Future longitudinal studies with objective measures of SHS exposure and physical fitness are warranted to confirm and extend our findings.

## 5. Conclusions

In conclusion, our study highlights the significant adverse effects of parental smoking on children’s lung function, BMI, and physical fitness. Children exposed to SHS exhibited lower lung function and physical fitness levels, underscoring the need for effective public health policies and interventions to reduce SHS exposure in homes. Promoting physical fitness in children can enhance overall health and serve as a protective measure against the respiratory and metabolic consequences of SHS exposure. By providing empirical evidence on the detrimental effects of SHS, our research contributes to the ongoing efforts to create smoke-free environments and promote healthy lifestyles, ultimately safeguarding the well-being of future generations.

## Figures and Tables

**Figure 1 children-11-01250-f001:**
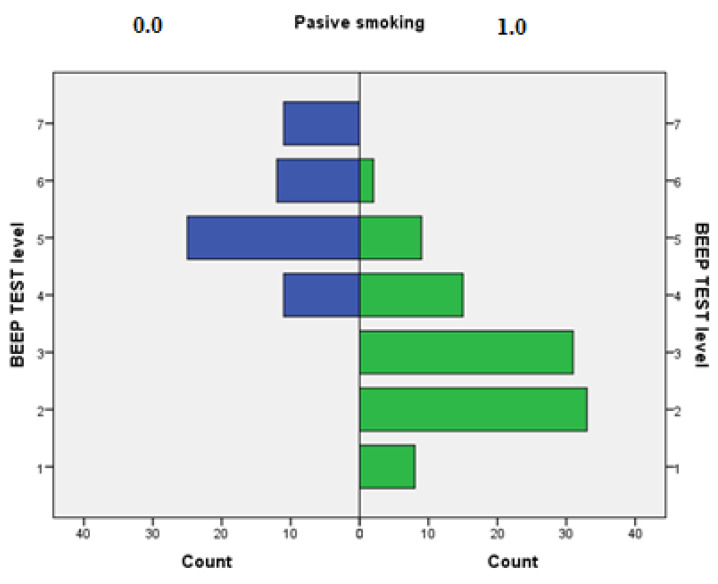
Association between passive smoking exposure and BEEP test levels.

**Table 1 children-11-01250-t001:** Baseline characteristics of the investigated population.

	Mean	Standard Deviation (SD)	
Age (years)	12.55	1.05	
	Median	Minimum	Maximum
BMI (z-score)	0.17	−2.71	2.33
FVC (z-score)	−0.05	−2.64	1.92
FEV1 (z-score)	−0.40	−2.69	2.13
FEV1/FVC (Tiffeneau) z-score	−0.20	−3.34	2.28
PEF (z-score)	−0.72	−3.64	2.96
		Count (N)	Percentage %
Sex	Boys	89	56.7%
	Girls	68	43.3%
Active sport	No	80	51.0%
	Yes	77	49.0%
Every day secondhand smoke exposure in a household	No	59	37.6%
	Yes	98	62.4%
Father–smoking status	No	82	52.2%
	Yes	75	47.8%
Mother–smoking status	No	78	49.7%
	Yes	79	50.3%
Parents–smoking status	No	59	37.6%
	Only one parent	42	26.7%
	Both mother and father	56	35.7%
Mother–smoking status during pregnancy	No	97	61.8%
	Yes	60	38.2%
BEEP TEST level	<P5 = very poor (level 1)	8	5.1%
	P5–P20 = poor (level 2)	33	21.0%
	P20–P40 = fair (level 3)	31	19.7%
	P40–P60 = average (level 4)	26	16.6%
	P60–P80 = good (level 5)	34	21.7%
	P80–P95 = very good (level 6)	14	8.9%
	>P95 = excellent (level 7)	11	7.0%

**Table 2 children-11-01250-t002:** Differences in lung function and BMI between children exposed and unexposed to SHS and active sport SHS-secondhand smoke.

		FVC (z-Score)		FEV1 (z-Score)		FEV1/FVC (Tiffeneau) z-Score		PEF (z-Score)		BMI (z-Score)	
		Median (Minimum, Maximum)	*p*-Value	Median (Minimum, Maximum)	*p*-Value	Median (Minimum, Maximum)	*p*-Value	Median (Minimum, Maximum)	*p*-Value	Median (Minimum, Maximum)	*p*-Value
Sex	male	−0.01 (−2.39. 1.32)	*p* = 0.694	−0.42 (−2.69. 1.71)	*p* = 0.347	−0.34 (−2.84. 2.28)	*p* = 0.572	−0.87 (−3.64. 2.53)	*p* = 0.134	0.17 (−2.71. 2.33)	*p* = 0.441
	female	−0.15 (−2.64. 1.92)		−0.19 (−2.54. 2.13)		0.005 (−3.34. 2.25)		−0.53 (−2.54. 2.96)		0.105 (−2.12. 1.96)	
Active in sport	No	−0.115 (−2.64. 1.92)	*p* = 0.178	−0.4 (−2.69. 2.13)	*p* = 0.112	−0.145 (−3.34. 2.28)	*p* = 0.368	−0.675 (−3.31. 2.04)	*p*= 0.384	0.12 (−2.71. 2.33)	*p* = 0.601
	Yes	0.06 (−1.94. 1.32)		−0.35 (−1.99. 1.85)		−0.2 (−2.25. 2.25)		−0.74 (−3.64. 2.96)		0.17 (−2.08. 1.93)	
Passive smoking exposure	No	0.47 (−0.65. 1.92)	*p* < 0.001	0.57 (−0.4. 2.13)	*p* < 0.001	0.51 (−0.91. 2.28)	*p* < 0.001	0.83 (−0.43. 2.96)	*p* < 0.001	−0.03 (−2.38. 1.92)	*p* = 0.018
	Yes	−0.395 (−2.64. 1.2)		−0.88 (−2.69. 1.22)		−0.73 (−3.34. 1.65)		−1.19 (−3.64. 0.53)		0.59 (−2.71. 2.33)	
Father–smoking status	No	0.265 (−2.64. 1.92)	*p* < 0.001	0.325 (−1.71. 2.13)	*p* < 0.001	0.34 (−2.84. 2.28)	*p* < 0.001	0.485 (−2.54. 2.96)	*p* < 0.001	0 (−2.71. 2.08)	*p* = 0.046
	Yes	−0.39 (−2.39. 1.2)		−0.89 (−2.69. 1.22)		−0.7 (−3.34. 1.65)		−1.17 (−3.64. 0.53)		0.57 (−2.08. 2.33)	
Mother–smoking status	No	0.32 (−2.11. 1.92)	*p* < 0.001	0.375 (−1.61. 2.13)	*p* < 0.001	0.395 (−2.25. 2.28)	*p* < 0.001	0.515 (−3.64. 2.96)	*p* < 0.001	−0.02 (−2.38. 1.92)	*p* = 0.011
	Yes	−0.55 (−2.64. 1.2)		−0.93 (−2.69. 0.37)		−0.75 (−3.34. 1.07)		−1.18 (−3.31. 0.04)		0.64 (−2.71. 2.33)	
Smoking exposure during pregnancy	No	0.17 (−2.11. 1.92)	*p* < 0.018	0.19 (−1.88. 2.13)	*p* < 0.046	0.1 (−2.25. 2.28)	*p* < 0.001	0.26 (−3.64. 2.96)	*p* < 0.016	0 (−2.71. 2.12)	*p* = 0.009
	Yes	−0.505 (−2.64. 1.2)		−0.885 (−2.69. −0.03)		−0.76 (−3.34. 1.07)		−1.21 (−3.31. −0.03)		0.76 (−2.12. 2.33)	

The correlation between pulmonary function (FVC, FEV1, FEV1/FVC, PEF) and BMI values was not statistically significant (*p* = 0.243, *p* = 0.176, *p* = 0.328, *p* = 0.207).

**Table 3 children-11-01250-t003:** Association of physical fitness (BEEP score) and lung function.

		FVC (z-Score)		FEV1 (z-Score)		FEV1/FVC (Tiffeneau) z-Score		PEF (z-Score)		BMI (z-Score)
		Median (Minimum, Maximum)	*p*	Median (Minimum, Maximum	*p*	Median (Minimum, Maximum	*p*	Median (Minimum, Maximum	*p*	Median (Minimum, Maximum
BEEP TEST level	1	−0.83 (−1.69. −0.24)	*p* < 0.001	−1.065 (−1.5. −0.08)	*p* < 0.001	−0.62 (−0.94. 1.02)	*p* < 0.001	−1.44 (−2.4. −0.21)	*p* < 0.001	1.655 (−0.29. 2.33)
	2	−0.32 (−2.64. 1.11)		−0.79 (−2.69. 1.22)		−0.68 (−2.84. 1.65)		−1.03 (−3.64. 0.53)		1.01 (−1.1. 2.08)
	3	−0.48 (−1.8. 1.2)		−0.93 (−2.54. −0.12)		−0.98 (−3.34. 0.99)		−1.24 (−3.31. −0.45)		0.16 (−2.71. 1.61)
	4	0.005 (−1.42. 1.32)		−0.3 (−1.81. 1.71)		−0.28 (−2.72. 1.57)		−0.3 (−2.53. 1.59)		−0.02 (−2.12. 1.92)
	5	0.25 (−1.85. 1.92)		0.4 (−1.99. 2.13)		0.12 (−1.2. 2.28)		0.825 (−2.9. 2.96)		0.135 (−2.08. 1.77)
	6	0.45 (−0.31. 1.32)		0.45 (−0.2. 1.25)		0.42 (−0.2. 1.7)		0.595 (−1.08. 2.04)		−0.375 (−2.38. 0.97)
	7	0.51 (−0.3. 1.3)		0.57 (−0.13. 1.56)		0.43 (−0.4. 1.29)		0.7 (−0.43. 2.53)		−0.61 (−1.37. 0.39)

FVC: forced vital capacity, FEV1: forced expiratory volume in 1 s, PEF: peak expiratory flow, BMI: body mass index.

## Data Availability

Data are available from the authors.

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
