# Peer review of "Secondhand Smoke Exposure and Its Impact on Pediatric Lung Function, Aerobic Fitness, and Body Mass: Evidence from a Cross-Sectional Study"

_children, 2024, doi:10.3390/children11101250_

Round 1
Reviewer 1 Report
Comments and Suggestions for Authors
The manuscript titled: Secondhand smoke exposure and its impact on pediatric lung function,aerobic fitness, and body mass: evidence from a cross-sectional study is very well prepared and interested for readers.
As Reviewer I would like to ask:
- if the SHS exposure was confirmed by biochemical tests, e.g. determination of cotinine concentration in the blood? The questionnaire is not a sufficient method for verification of smoke exposure.
- in lines: 84-85 and 93-94 are included the same information (please rewritten)
- in Statistical analyses there are no information dealing with the tests used to check normal distribution of variables. On what basis the non-parametric tests used in these analyses were chosen?
- Table 1, Line “Parents smoking status: “Mother or father” is used twice. It is not clear.
Reviewer 2 Report
Comments and Suggestions for Authors
Dear authors, the topic of the study is well-chosen and will be of interest to the reader. However, the research has some shortcomings.
The research group is 10-14 years old. There is a high possibility of biased results when comparing these age groups. For example, a 14-year-old child with SHS may show better results than a 10-year-old child without SHS due to development. The same is valid for gender. There are expected to be differences between genders in adolescence. Shouldn't you have compared gender and age here?
Standardization was done with Z score, but as I just mentioned, there are gender and age differences. A total of 157 people participated in the study, and parametric analysis can be done for numbers above 30. A t-test should be performed instead of the Mann-Whitney u test. In addition, correlation analysis is mentioned in the statistical analysis section, but no results about correlation are presented.
Another issue is that there were children doing sports in the study. Shouldn't the values of children who do sports be higher?
Was the SHS evaluation made according to the parents? Were the children asked?
Comments on the Quality of English LanguageMinor editing of English language required.
Reviewer 3 Report
Comments and Suggestions for Authors
As an active opponent of any kind of smoking, as a parent, athlete, and scientist, I am glad that the authors decided to investigate this topic. This also tells you that the work has a huge readership potential, both for ordinary people and for specialists in the field that the work deals with. The work is well methodologically laid out. In the introduction of the paper, the authors correctly explain the terms related to the problem of the paper, using adequate references. Speaking of them, they were used throughout the work correctly and relied on recent research. The sample of respondents, which was derived from the number of residents of the local community (9000), is appropriate (157) for this type of cross-sectional study. Researchers often ignore this fact and go in the wrong direction, especially when concluding, which is not the case in this paper. All instruments for data collection are validated and reliable and are used exactly for what they are intended for, so you should not doubt the reliability of the data obtained. The obtained results are nicely packaged in tables and figures, and a well-developed discussion supported by similar research enabled the authors to draw correct conclusions. In this sense, I consider the following suggestions to be minor, but still important for the overall quality of the work, and I suggest that the work be accepted with minor corrections.
Line 112 Information on exposure to cigarette smoke was collected using a self-created questionnaire. Please describe the type and number of questions in more detail, as in further work I see that there are more than two statements (smokers and non-smokers).
Line 118 Weight and height were measured using a digital electronic weighing scale (range 1-118150 kg) and a digital stadiometer (range 70-205 cm). Considering that this is a scientific work, it is essential to provide precise facts, so here it is necessary to give the brand and type of scales for measuring weight and the apparatus used for height (as you stated for the spirometer). Scales and anthropometers must be calibrated and accurate.
In Table 1.
the last part
Parents smoking status
The last line is not entirely clear to me, and it doesn't fit with the results three lines above
No 97 61.8% do not fit me. Additional explanation if possible
Line 316 While our study provides valuable insights, it is not without limitations.
For the clarity of the paper, put this part in the chapter Limitations of the study and recommendations for future research
Round 2
Reviewer 1 Report
Comments and Suggestions for Authors
I recommend it for publication in the present form.
Reviewer 2 Report
Comments and Suggestions for Authors
Dear author;
Thank you for your reply and corrections. Use "mean" instead of "median" in Tables 1, 2 and 3.
Comments on the Quality of English LanguageMinor editing of English language required.